# Wear Diagnostics of the Thrust Bearing of NK-33 Turbo-Pump Unit on the Basis of Single-Coil Eddy Current Sensors

**DOI:** 10.3390/s21103463

**Published:** 2021-05-16

**Authors:** Viktor Belosludtsev, Sergey Borovik, Valeriy Danilchenko, Yuriy Sekisov

**Affiliations:** 1Samara Federal Research Scientific Center RAS, Institute for the Control of Complex Systems RAS, 443020 Samara, Russia; iccs@iccs.ru (V.B.); vdan26@mail.ru (V.D.); sekisov@iccs.ru (Y.S.); 2Public Joint-Stock Company, “UEC-Kuznetsov”, 443009 Samara, Russia

**Keywords:** liquid-propellant jet engine, turbo-pump unit, thrust bearing, wear diagnostics, single-coil eddy cur-rent sensor, laboratory prototype

## Abstract

The problem of early wear diagnostics of the combined journal-and-thrust bearing of the turbo-pump unit (TPU) of the liquid-propellant rocket engine NK-33 is considered. A feature of the problem is the significant restriction on modifications of the power plant’s design. The original solution based on replacing the standard induction sensors of the turbo-pump rotational speed currently used in TPU by single-coil eddy current sensors (SCECS) with sensitive elements in the form of a segment of a linear conductor is proposed. The SCECS provide the monitoring of the axial displacement of the shaft in the thrust bearing, which characterizes the state of the unit and increases with the bearing wear. The function of the TPU shaft’s rotational speed measuring also remains. The article describes the proposed approach as well as a laboratory prototype of the system for early detection of the TPU thrust bearing’s wear. The results of the prototype research that confirm the feasibility of the proposed approach are analyzed.

## 1. Introduction

The liquid-propellant rocket engine (LPRE) NK-33 was developed by academician N.D. Kuznetsov’s construction office for the lunar program N1-L3. It is the world’s first single-chamber rocket engine with a thrust of more than 100 tnf that is made on a closed circuit, based on oxygen-kerosene components, has a multiple launch and is reusable [1]. Even though the engine was created in the late 60 s of the last century it is still outstanding in some of its performance characteristics and remains the world priority in terms of reliability and technical excellence. It is stated in [2], with the reference to the representatives of the American rocket and missile propulsion manufacturer “Aerojet Rocketdyne”, that “NK-33 is the most reliable of all existing engines operating on oxygen and kerosene and it demonstrates the maximum thrust-to-weight ratio.” This was the reason why one of the modifications of the NK-33 (AJ-26) was used on the Antares launch vehicles. Its inaugural test launch with two AJ-26 rocket engines took place on 21 April 2013 [3]. The projects of new Russian launch vehicles “Yamal”, “Aurora”, and “Soyuz-2-3” with modified NK-33 engines are considered too. It is expected that NK-33 engine will significantly increase the useful payloads delivered into the orbit [2].

The turbo-pump unit (TPU) is one of the main components of the LPRE that provides the liquid rocket fuel flow. The fuel is in close proximity to the oxidizer in TPU and the development of the destruction processes in the unit elements could have disastrous consequences both for the power plant and for the launch vehicle in general [4]. Therefore, the reliable operation of the TPU and the early detection of the breakdowns of the TPU’s structural elements that can lead to an engine explosion are the tasks of vital importance.

In turn, the combined journal-and-thrust bearing (CJTB) is a critical component of the TPU which has significant mechanical loads and whose accelerated wear is accompanied by the destruction of the entire power plant. TPUs of modern modifications of the NK-33 have an efficient automatic unit for the CJTB unloading from axial forces. However, according to [4] the failure and destruction of the bearing unit are still possible. The lack of means for CJTB wear monitoring does not allow to detect the beginning of the bearing’s destruction and, therefore, does not make it possible to prevent the occurrence of an emergency.

The various approaches for the diagnosis of the bearing mount assemblies state in power plants are currently known. The detection of the metal wear particles in the lubrication system of the bearing assembly [5,6,7,8,9], the acoustic or vibration signal analysis [10,11], the temperature measurement [12,13] or their combination [14] are the most popular methods among those used in the aerospace applications. At the same time considering the strict requirement from the LPRE developers of the inadmissibility of any changes in the TPU body, the application of these methods is not possible, because they all require modifying the construction of the TPU to install the appropriate sensors.

In [15], a displacement monitoring method for the non-contact fault diagnosis and preventive maintenance of axle box bearings of rail vehicles is proposed. It is based on computer vision to monitor the vertical displacement of the bearing under simulated real working conditions. Although the method does not involve a change in the design of the monitored unit, it also cannot be implemented, because it requires the installation of the portable camera on the central axis on the side of the bearing assembly. What is more, this is not possible on the engine under consideration.

The article proposes a solution to the problem of the CJTB early wear diagnostics based on replacing the standard induction sensors of the turbo-pump rotational speed by single-coil eddy current sensors (SCECS) with sensitive elements (SE) in the form of a segment of a linear conductor [5,16,17,18] that constitute a separate and independent branch among the eddy current sensors used in aerospace engine building [19,20,21,22]. The SCECS provide the monitoring of the axial displacement of the shaft in the CJTB, which characterizes the bearing wear. In this case, the function of the rotational speed measuring of the TPU shaft is preserved, and no modifications of the power plant’s design are made. The last factor, as noted above, is a fundamental condition from the LPRE developers.

The article has four sections, in addition to the introduction and the conclusion. The sections consistently provide the explanation of the proposed approach for the early diagnostics of the CJTB wear of the TPU of NK-33, the description of the design and functional features of the SCECS with SE in the form of a segment of a linear conductor and its modification intended for installation on a TPU’s body as well as the description of the principles of converting the SCECS’s output parameter (inductance) in the measuring circuit (MC) and further processing of the SCECS’ information signals to determine the TPU shaft’s rotational speed and the axial displacement of the shaft in the thrust bearing, which characterizes the state of the monitored unit. The results of the feasibility testing of the proposed approach obtained on a laboratory prototype of the monitoring and diagnostic system are given too.

## 2. Evaluation of the CJTB’s Wear Based on Monitoring the Axial Displacement of the TPU’s Rotor

The main purpose of the CJTB in TPU is to ensure the position of the turbo pump rotor relative to the stator, to take the axial pressure of the rotor and to prevent it from the moving in the direction of its own axis. The significant axial loads acting on the TPU’s rotor during the power plant operation can lead to the destruction of the inner surface of the thrust bearing and its failure even though the special unloading devices and accessories are used. The shaft of the TPU’s rotor is dis-placed in the direction of the axial force with the abrasion of the CJTB’s internal surface. The magnitude of this displacement can inform about the degree of unit’s wear. The NK-33 engine and its elements have a complete design. That is why the requirement to save all the existing LPRE monitored parameters and to avoid any changes in the design of the engine body including through the installing additional sensors for rotor’s axial movement monitoring is a mandatory condition of the modernization of the system for monitoring and control the dangerous states of the engine.

The methods for measuring of radial and axial displacements of the power plants’ structural elements based on the use of SCECS with SE in the form of a segment of a linear conductor are currently known [16,23,24]. They provide a reliable functioning in the extreme conditions typical to aerospace application. It is shown in [25] that SCECS can also be used to measure the rotational speeds and angular accelerations of turbomachine rotors, and this carries out the second prerequisite in terms of maintaining the existing monitored parameters of the rocket engine. Obviously, these methods and technical means that implement the methods can be also used to solve the problem of early wear diagnostics of the CJTB of NK-33 TPU.

Considering the strict requirements of the inadmissibility of any changes in the TPU body the authors propose to modify the standard rotor’s speed measuring unit (Figure 1a) by replacing the inductive IS-445 RPM sensors (RPMS_1_ and RPMS_2_) with specially designed SCECS that are like IS-445 in size and mounting dimensions. The structure of the modified unit for measuring the TPU rotor’s speed and diagnosing the CJTB wear is shown in Figure 1b. Considering the specifics of the SCECS functioning the measuring disk made of magnetic steel should be also replaced with a non-magnetic one while remaining its strength, weight, and size.

The SCECS_1_ and SCECS_2_ are mounted in the same holes as the corresponding RPMS_1_ and RPMS_2_ in the standard rotor’s speed measuring unit (Figure 1a). In turn, the RPMS signal processing unit (Figure 1a) is replaced by the measuring converter (Figure 1b) which contains the individual MCs for each SCECS, the individual generators of the rotor’s rotational speed signals (RSG_1_ and RSG_2_), and the generator of the shaft’s axial displacement signals (ADG) common to both sensors.

Sensitive elements of each SCECS_1_ and SCECS_2_ are connected to their own MC_1_ and MC_2_ through current leads and matching transformers (MTs). The MCs convert the sensors’ natural output parameters (inductances) into the analog signals in the form of voltages *U*_1_ and *U*_2_. The following transformations of *U*_1_ and *U*_2_ are carried out in RSG_1_ and RSG_2_ which generate the output signals in the form of voltage pulses with amplitudes and frequencies (*f*_1_ and *f*_2_), similar to the output signals of the standard rotor’s speed measuring unit (Figure 1a).

The voltages *U*_1_ and *U*_2_ come in parallel at the ADG input. The ADG contains a multi-channel analog-to-digital converter (ADC) and a computer (microcontroller with all necessary periphery). The voltages *U*_1_ and *U*_2_ are converted in ADG into the digital codes and the shaft’s axial displacement in the CJTB (*x*) that characterizes the bearing wear is calculated. The pre-obtained set of calibration characteristics of the measuring channels with SCECS_1_ and SCECS_2_ in the form of dependencies of digital codes on the corresponding ADC outputs on axial displacement of the shaft and radial clearance (RC) between sensors’ SE and the protrusion of the measuring disk is used to compute *x*. The value of the shaft’s axial displacement (*x*) in the form of a digital code is transferred further to the monitoring unit for comparison with the maximum permissible limits and for appropriate decision making.

Thus, the proposed modification of the TPU rotor’s speed control unit contains two independent channels for the rotation speed measuring that completely simulate the similar channels of the standard monitoring system and do not require any changes in the existing signal processing algorithms. The unit also contains the additional channel for measuring the axial displacement of the shaft in the CJTB that expands the functional capabilities of the unit, makes it possible to evaluate the bearing wear and provides the formation of the signal for the emergency shutdown of the engine when necessary. The operating features of the monitoring system’s original functional elements are further considered.

## 3. Single-Coil Eddy Current Sensors with Sensitive Element in the Form of a Segment of a Linear Conductor and Their Location on TPU Body

The schematic image of a SCECS, its electrical configuration and the results of electromagnetic interaction of the SE of SCECS with the protrusion tip of a measuring disk are shown in Figure 2.

The sensor consists of three elements (Figure 2a): a matching transformer (MT), “non-inductive” current leads made as closely spaced and isolated coaxial cylinders and a SE in the form of a conductor segment. The current leads and the SE form the secondary MT winding. The primary MT winding is included in the MC with pulsed supply which implements the method of the first derivative [16].

The value of the SCECS information parameter—its equivalent inductance (*L_eq_*), depends on the actual position of the protrusion on the measuring disk in relation to the sensor’s SE. Let us assume that the MT does not distort the front edge of the supply voltage of a rectangular shape, which excites the *i*_SE_(*t*) current in the SE circuit (Figure 2b). If the protrusion of the measuring disk is outside the SCECS sensitivity zone and the influence of eddy currents in the SE associated with the magnetic field caused by the current *i*_SE_(*t*) can be neglected, the inductance *L_eq_* = *L*_0_ [16], where *L*_0_ is the self-inductance of the SE, recalculated to the primary winding of the MT (Figure 2c).

With the approach of the protrusion of the measuring disk to the SE, the eddy currents appear in the protrusion under the action of a magnetic field created by *i*_SE_(*t*) and time-varying current *i*_p_(*t*) appears in the contour that imitates the protrusion (Figure 2b). The current *i*_p_(*t*) affects the resulting magnetic field and this leads to changes of the current *i*_SE_(*t*) and, as a result, to changes (decrease) of the SCECS equivalent inductance *L_eq_*. It is shown in [16] that at the beginning of the transition process at *t*
→ 0, the SCECS equivalent inductance depends only on the mutual inductance of the currents’ contours *i*_SE_ and *i*_p_, i.e., depends on the distance between the protrusion and the sensor’s SE. The equivalent inductance of the SCECS will be minimal (*L_eq,min_*) when the geometric center (g.c.) of the protrusion’s tip aligns with g.c. of the SE (rotor rotational angle Ψ_0_ on Figure 2b). Thus, *L_eq,min_* is taken as the information value. The exact meaning of *L_eq,min_*(Ψ_0_) is determined by the overlap between the SE and the protrusion of the measuring disc (the axial displacement of the protrusion) and by the radial clearance (RC) between the protrusion’s tip and the SE.

It should be noted that the use of SCECS for CJTB wear diagnostics requires the solution of two contradictory problems. On the one hand, the protrusion axial displacements associated with the wear of the bearing elements are small. That is why the maximum sensitivity of the sensor to the TPU’s rotor (measuring disk) displacements is required. On the other hand, the working range of the measured displacements should be wide enough because the dimensional chain in the direction of the rotor axis in addition to shaft displacements due to the CJTB wear contains other components related to the initial axial clearances in the bearing, tolerances on its dimensions, etc. Moreover, their value can significantly exceed displacements caused by the CJTB wear. To meet the requirements in the working range of the measured axial displacements the length of the SCECS’s SE is selected equal to the length of a protrusion of the measuring disk. The maximum possible sensitivity of the SCECS and the linearity of its characteristics are achieved due to the angular orientation of the sensor’s SE parallel to the protrusion tip. Figure 3 provides a schematic representation of the SCECS’s SE relative position with respect to the protrusion of the measuring disc. Here the selected Cartesian coordinate system *OXYZ* is shown. The origin of the coordinate system is attached to the SCECS’s SE, axis *X* characterizes the axial displacement of the measuring disc, axis *Y*—the RC between the SE and the protrusion, and axis *Z*—the protrusion’s linear displacement in the direction of the rotation of the measuring disc.

Although *L_eq,min_*(*x*) is of interest only for the evaluation of CJTB wear, the RC between the protrusion of the measuring disc and SCECS’s SE (*y*-coordinate) must be taken into account. The RC changes in the operation of TPU under the influence of elastic and temperature deformations of the measuring disc and is inherently an interfering factor. Taking into account the above reasoning, at a time relevant to the angle position of the measuring disc Ψ_0_ we can write:(1)$$L_{eq,min}\left(x,y\right)=L_{0}-\Delta L=L_{0}-\Delta L_{max}\left(\frac{1}{1+K_{Y}y}\right)\left(1-\frac{x}{b_{SE}}\right),$$
where Δ*L_max_* is the extreme change of sensor’s equivalent inductance at zero RC and full overlap of a protrusion by SE, *K_Y_* is the coefficient that characterizes the sensor’s sensitivity to the RC when SE is completely overlapped by a protrusion, and *b_SE_* is SE’s length. The exact meanings of *K_Y_* and Δ*L_max_* are determined experimentally during the SCECS’s calibration process on specialized benches [16].

It is obvious that expression (1) is not solvable for *x* if RC (coordinate *y*) is unknown. This means that the desired axial displacement of the shaft in CJTB cannot be determined by a single sensor. In this case the methods based on the clusters (groups of the identical sensors) of SCECS can be used to measure the multi-dimensional displacements of power plants’ structural elements [23,26]. The number of sensors in the cluster is determined by the number of measured coordinates of displacement. According to the methods, the inductances of the primary MT windings of the SCECS (information parameters) are fixed at a specific time when the controlled elements pass the selected reference system. Further the desired coordinates are calculated by joined processing of measurement data from each sensor using the pre-obtained calibration characteristics.

Two mounting holes for standard RPM sensors in TPU body make it possible to fully implement a variant of the method for measuring RC and axial displacements of power plants’ structural elements with distributed cluster of SCECS [24]. SCECS are placed into the mounting seats on the TPU body intended for standard RPM sensors. As already mentioned, SE of both SCECS are oriented parallel to the long side of the protrusion’s tip on the measuring disc. The g.c. of the SE of SCECS_1_ shifts relative to the g.c. of the mounting hole (sensor) by a value equal to $-\frac{1}{2}\Delta x$ and the g.c. of the SE of SCECS2 shifts by a value equal to $+\frac{1}{2}\Delta x$, where Δ*x* is the range of possible axial displacements of the TPU rotor. In this case the axial displacement of the shaft in CJTB can be measured in all range of its possible variation. The last condition requires to modify the design of the SCECS by changing the position of its SE’s g.c. across a vertical axis of the sensor. The possible variant of a new design of the SCECS (the end side of the sensor) and the placement of two SCECS in mounting holes on the TPU body are shown in Figure 4.

## 4. Measuring Circuit and Conversion of SCECS’s Information Signals

The equivalent inductances of both SCECS are converted into the corresponding voltages in individual MC which were constructed according to the differential scheme and implemented the method of the first derivative. The method provides for fixation of the derivative of the current in the MT primary winding of SCECS at the moment of supplying a power pulse to the MC with the sensor. At this time, the changes in the equivalent inductance of the SCECS associated with the passage of the protrusion of the measuring disk under the sensor’s SE are of the largest magnitude. The differential connection of SCECS allows to allocate only the informative part of the equivalent inductance’s changes (Δ*L*) in the MC’s analog output signal and to exclude the non-informative component *L*_0_.

Traditionally, such MC are built on the basis of the Blumlein Bridge [16] or nonequilibrium bridge with operational amplifier in differentiation mode [23,26]. Furthermore, one leg of the bridge includes the working SCECS, and the second one—either a similar SCECS performing the compensatory functions or its simulator in the form of an inductance coil, the value of which is equal to its own inductance of the working SCECS *L*_0_ [27]. In the present case the variant of the differential MC with SCECS simulator (SCECSS) is only applicable because of the impossibility of location additional sensors on the TPU body.

The amplitude value of the output voltage of the MC with SCECS and SCECSS, can be defined as:(2)$$U_{out}=E\frac{\Delta L}{L_{0}}K_{S},$$
where *E* is the voltage of the MC power supply and *K_s_* is the coefficient that characterizes the MC sensitivity.

Considering (1), Equation (2) can be rewritten for the moment when the protrusion of the measuring disk is exactly below the SE of SCECS (the rotational angle of the disk is Ψ_0_, Figure 3) as:(3)$$U_{out}=E\frac{\Delta L_{max}}{L_{0}}K_{S}\left(\frac{1}{1+K_{Y}y}\right)\left(1-\frac{x}{b_{SE}}\right).$$

Figure 5 schematically shows the location of the protrusion of the measuring disk relative to the SE of SCECS_1_ and SE of SCECS_2_ at the moments when the axial displacement of the TPU rotor’s shaft is negative (Figure 5a), absent (Figure 5b) and positive (Figure 5c). The origins of local coordinate systems *O*_1_*X*_1_*Y*_1_ and *O*_2_*X*_2_*Y*_2_ are attached to the g.c. of SCECS_1_ and g.c. of SCECS_2_, respectively, the axis *Z* is not shown in Figure 5. To simplify the further reasoning, it is assumed that all the parameters of SCECS_1_ and SCECS_2_ as well as the parameters of MC_1_ and MC_2_ are identical.

If the axial displacement of the shaft is absent (Figure 5b), then the protrusion of the measuring disk overlaps the SE of both SCECS in equal proportion and if the RC is constant, the voltage at the outputs of MC_1_ and MC_2_ will be equal:(4)$$U_{out1}=U_{out2}=0.5E\frac{\Delta L_{max}}{L_{0}}K_{S}\left(\frac{1}{1+K_{Y}y}\right).$$

The effect of shaft’s axial displacement is that the measuring disk with a protrusion will be closer to one of the SCECS. Thus, the overlapping area of the SE of the SCECS by the protrusion will increase, and the overlapping area of the SE of the second sensor by the protrusion, respectively, will proportionally decrease:(5)$$\left\{ \begin{array}{l} U_{out1}=E\frac{\Delta L_{max}}{L_{0}}K_{S}\left(\frac{1}{1+K_{Y}y}\right)\left(1-\frac{b_{SE}/2\mp x}{b_{SE}}\right),\\ U_{out2}=E\frac{\Delta L_{max}}{L_{0}}K_{S}\left(\frac{1}{1+K_{Y}y}\right)\left(1-\frac{b_{SE}/2\pm x}{b_{SE}}\right). \end{array}\right.$$

Figure 6 demonstrates the voltages diagrams of the output signals of MC_1_ and MC_2_ corresponding to the same axial displacements of the measuring disk relative to the SE of SCECS_1_ and SE of SCECS_2_ shown in Figure 5. It is assumed that RC remains unchanged during the measurement and the disk has two identical protrusions located at diametrically opposite points along its generatrix.

Further processing of the SCECS’ information signals is carried out on a digital level. It involves the analog-to-digital conversion of *U_out_*_1_ and *U_out_*_2_ voltages into the numeric codes *C*_1_ and *C*_2_ and the computation of the desired shaft displacement in CJTB which is calculated on the basis of the pre-obtained calibration characteristics of the measuring channels in the form of the dependencies of the digital codes corresponding to the equivalence inductions of the SCECS (voltages at the outputs of MC_1_ and MC_2_), RC (coordinate *y*), and the axial displacement of the measuring disk (coordinate *x*):(6)$$\left\{ \begin{array}{l} C_{1}=f_{1}\left(x,y\right)\\ C_{2}=f_{2}\left(x,y\right) \end{array}\right..$$

The solution of the system (6) is preceded by a preliminary approximation of the set of calibration characteristics by the polynomial functions of two variables or by line-segment (line-polynomial) interpolation. Generally, if the calibration characteristics are monotonic in a specified range of variation of the *x*,*y*-coordinates, the algorithm based on the Newton method is used for computation of the desired coordinates of the displacement of the measuring disk [23]. In the case when the calibration characteristics and their approximating functions become nonmonotonic, the system of equations (6) cannot be solved by these algorithms. An algorithm that can be used for solving the system (6), in which there is no requirement of monotonicity of the functions **f**_1_(*x*, *y*) and **f**_2_(*x*, *y*) was described in [26].

In turn, the time interval between the extreme values of the adjacent bell-shaped pulses corresponding to the passage of the protrusion of the measuring disk under the SE of the same SCECS is calculated to determine the rotation speed of the disk and to generate the corresponding signals for the TPU’s regular monitoring system. It should be noted that the method for determining the rotation speed and angular acceleration of a disk with discrete generating (blade wheel, gear, disc with protrusions, etc.) is discussed in detail in [25]. For that reason, it is not given in current section.

## 5. Pilot Testing of the Proposed Approach Feasibility

The laboratory bench was developed for the testing the feasibility of the method proposed in the article and for debugging the algorithms for detecting the axial displacements of TPU rotor’s shaft in the CJTB. The external view of the bench is shown in Figure 7a. It includes a TPU simulator containing a stator (2) with installed SCECS_1_ and SCECS_2_ (1), a simulator of the measuring disk (3) made of stainless non-magnetic steel with two protrusions, a block of measuring transducers (5) containing MC_1_ and MC_2_, a signal processing unit (6), and an external module E14-440 by L-Card (4) with a 14-bit ADC for measurement information entry into a PC [28].

To perform the experiments, we used a specially designed SCECS. The diameter of the sensor’s external current lead was about 20 mm and its SE length was about 10 mm. The g.c. of the SE has the offset about 5 mm relative to the sensor’s central axis. The Figure 7b shows the installation of SCECS on the stator of the laboratory bench.

The initial RC between the protrusion and the SE of SCECS was set to 0.3 mm. The range of the axial displacements of the disc was limited to ±2 mm. The control of the RC and axial displacements was carried out by dial indicators with a resolution 0.01 mm. The measuring disk was rotated by a DC motor with controlled rotation speed. In the experiments the rotation speed of the disk was set to 6500 rpm and was limited to 8000 rpm according to the safety requirements.

Figure 8 displays the screenshots of the program for scanning the measuring channels of the diagnostic system’s prototype. The results were obtained for the initial conditions specified above during the rotation of the measuring disc. The axial displacement of the disk relative to the g.c. of the SCECS was taken as 0 mm (Figure 8a), −2.0 mm (Figure 8b), and +2 mm (Figure 8c). The curve 1 (red) on the graphs corresponds to the protrusion’s passage under the SE of SCECS_1_ and curve 2 (green) corresponds to the protrusion’s passage under the SE of SCECS_2_.

As it can be seen from the diagrams in Figure 8, the amplitude values of the pulses corresponding to the passage of the measuring disk protrusions through the SCECS_1_ and SCECS_2_ sensitivity zones in the absence of axial displacements of the disk are approximately the same and equal to about 3000 code units. The axial displacement of the measuring disk to the far-left position (−2.0 mm) increases the amplitude of the pulses when the protrusions pass the SE of SCECS_1_ approximately equal to 4550 code units and simultaneously decreases the amplitude of the pulses associated with the passage of the same protrusions of the SE of SCECS_2_ approximately equal to 1700 code units. The axial displacement of the measuring disk to the far-right position (+2.0 mm) changes the signals at the output of the measuring channels with SCECS_1_ and SCECS_2_ to a reverse value.

Using the same laboratory bench, but in static, the calibration characteristics of the measuring channels with SCECS were experimentally obtained in the ranges of variation of the *x* coordinate from −2 to +2 mm and of the *y* coordinate from 0.3 to 0.5 mm. They were approximated by the polynomial function of two variables:(7)$$C_{k}=\sum_{i=0}^{I}\sum_{j=0}^{J}a_{ij}x^{i}y^{j},$$
where *k* is the SCECS’s number (1 or 2), *I* is the polynomial order by *x*, *J* is the polynomial order by *y*.

Considering (6) and (7), the calibration characteristics for the case under review were presented as polynomials
(8)$$\left\{ \begin{array}{l} C_{1}=6.9\cdot 10^{3}-1.3\cdot 10^{4}y-1.35\cdot 10^{3}x+2\cdot 10^{3}xy-37.5x^{2}+125x^{2}y\\ C_{2}=5.7\cdot 10^{3}-9\cdot 10^{3}y+1.19\cdot 10^{3}x-1.63\cdot 10^{3}xy+81.25x^{2}-187.5x^{2}y \end{array}\right..$$

The polynomials are monotonic in specified range of RCs and axial displacements of the measuring disk and the well-known algorithms based on the Newton method [23] can be used to compute the *x*,*y*-coordinates of the disk’s protrusion displacement. The axial displacements of the measuring disk calculated in bench working mode (during the disk rotation) were in the range −0.14…+0.02 mm when the protrusion was approximately at the equal distance from the g.c. of SCECS’ SEs (this corresponds to Figure 8a). For the operating modes of the laboratory bench that correspond to Figure 8b,c the calculated axial displacements of the disc were in the range −2.0…−2.1 mm and +1.7…+1.85 mm, respectively. These results fit to the initial data. The variation of the results within ±0.08 mm may be related to the uncertainty of the disk’s orientation in the rotation plane when it was mounted on the shaft and to the possible beats of the rotor of the DC motor in the axial direction.

The angular speed of the measuring disk rotation was determined by the distance between the extreme values of the neighboring pulses corresponding to the passage of the protrusions under the SE of the same SCECS (SCECS_1_ or SCECS_2_). This distance equals 370 code units for diagrams in Figure 8. Considering the ADC sampling rate 200 kHz and the presence of two protrusions on the measuring disk the angular speed of rotation estimated 6480 rpm. The result corresponds to the initial data.

Thus, it can be argued that laboratory tests fully confirm the feasibility of the proposed in the article approach to determining the axial displacements of the shaft in CJTB, which characterize the wear of the bearing unit and can be the source of a primary information for early diagnosis of its condition. The measuring channels of the system’s prototype are characterized by sufficient sensitivity to the measured parameters. The sensitivity of both channels to the axial displacements of the measuring disk was more than 1400 code units per mm in experiments for RC 0.3 mm. The sensitivity of the measuring channels to the axial displacements decreases as the RC increases, and for RC equal to 0.5 mm it is about 300 code units per mm. However, even in this case it remained at a sufficient level to determine the desired diagnostic parameter with acceptable accuracy. It should also be noted that the actual operating conditions of the TPU are characterized by higher rotational speeds of the measuring disk and higher temperatures in the measuring zone. That is why the RC between the protrusions and stator shell is likely to be less than in laboratory tests due to the larger elastic and temperature stresses of the disk.

## 6. Conclusions

The article considers an approach to the early diagnosis of the CJTB wear of the TPU of NK-33 based on the analysis of the axial displacement of the shaft in a bearing. The proposed modernization of the system for monitoring and control the dangerous states of the NK-33 engine involves the replacement of standard RPM-sensors IS-445 with the original SCECS with the SE in the form of a segment of a linear conductor. As a result, the TPU design does not require any change. The function of the TPU shaft’s rotational speed measuring that exists in the standard monitoring and control unit also remains.

The proposed approach was tested in laboratory conditions on a specialized bench with a real measuring disk. The experiments fully confirmed the feasibility of the proposed approach to deter-mining the axial displacements of the shaft in CJTB, which characterize the wear of the bearing unit and can be the source of a primary information for early diagnosis of its condition. The sensitivity to the axial displacement of the measuring disk was estimated, too. For RC 0.3 mm, it was about 1400 code units per mm, thus ensuring the identification of the desired diagnostic parameter with acceptable accuracy.

## Figures and Tables

**Figure 1 sensors-21-03463-f001:**
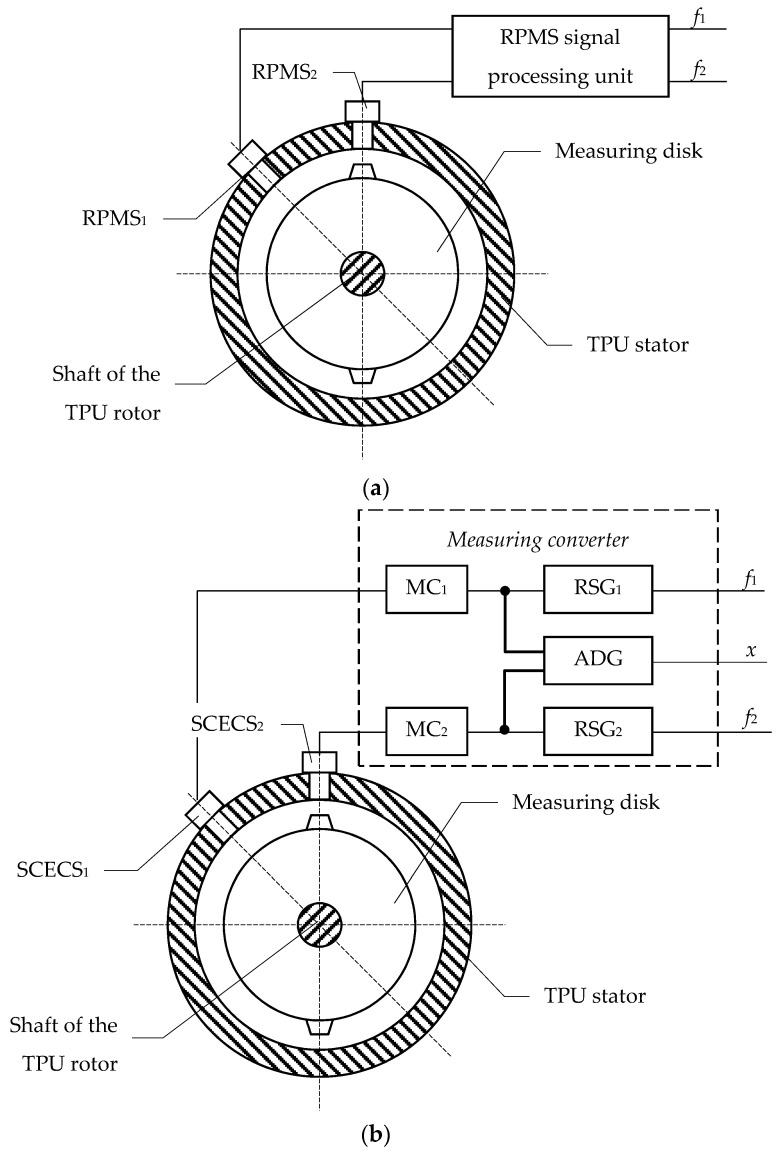
Existing TPU rotor’s speed measurement unit (**a**) and its modification for CJTB wear diagnostics (**b**).

**Figure 2 sensors-21-03463-f002:**
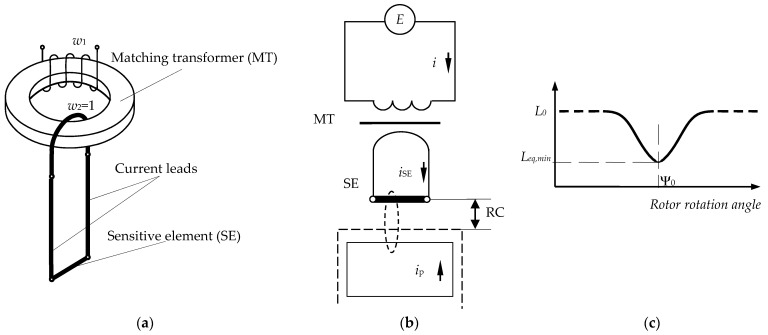
Schematic image of a SCECS (**a**), its electrical configuration (**b**), and the changes in equivalent inductance of sensor’s SE when the protrusion tip of the measuring disk passes the SCECS sensitivity zone (**c**).

**Figure 3 sensors-21-03463-f003:**
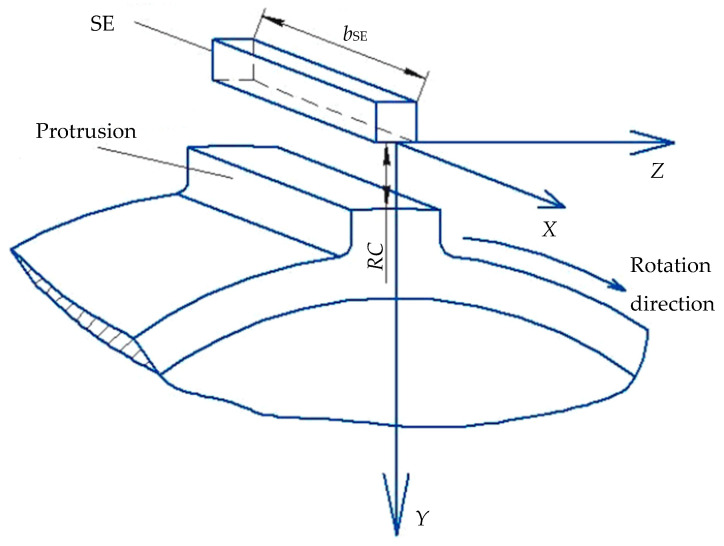
SCECS’s SE relative position with respect to the protrusion of the measuring disc.

**Figure 4 sensors-21-03463-f004:**
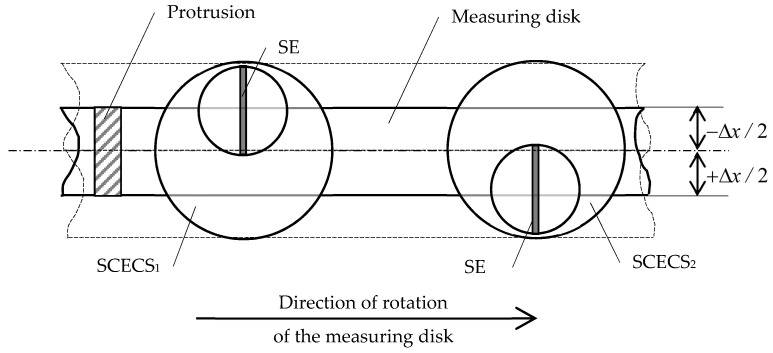
Placement of two SCECS on the TPU body and the orientation of sensors’ SE.

**Figure 5 sensors-21-03463-f005:**
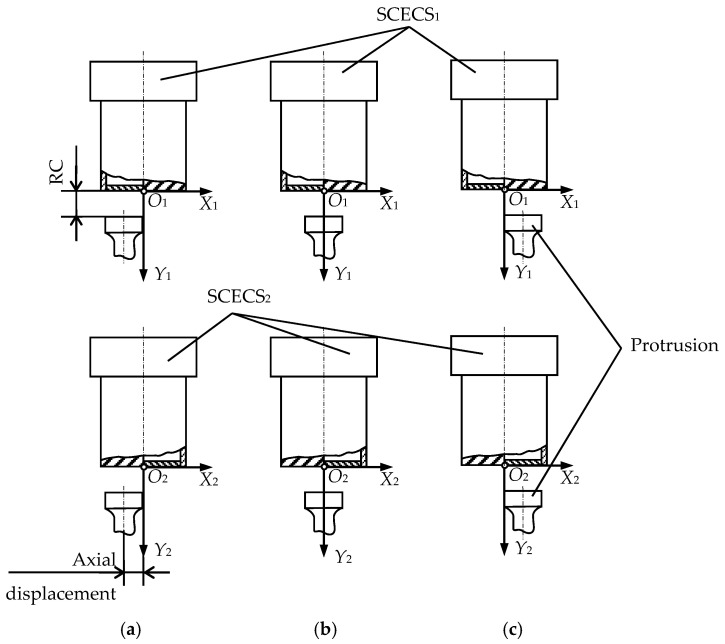
Location of the protrusion of the measuring disk relative to the SE of SCECS_1_ and SE of SCECS_2_ at the moments when the axial displacement of the TPU rotor’s shaft is negative (**a**), absent (**b**), and positive (**c**).

**Figure 6 sensors-21-03463-f006:**
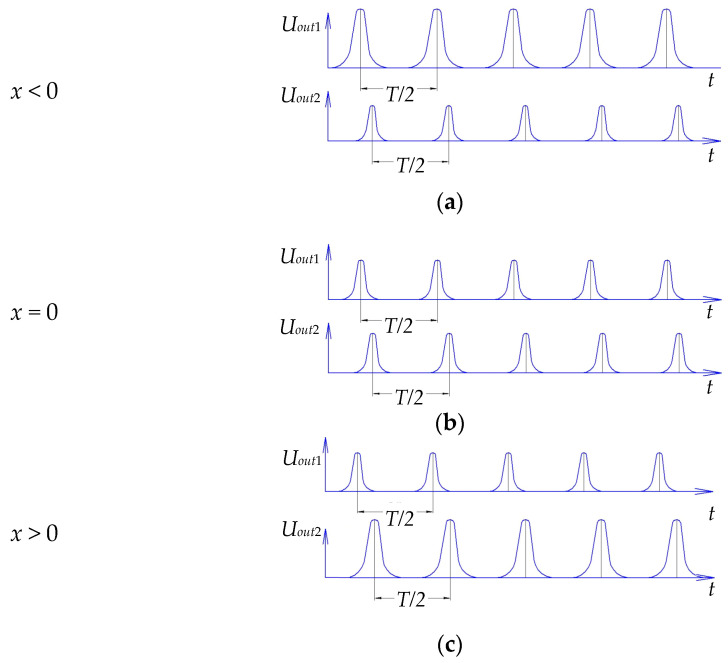
Time diagrams of changes in the output signals of the MC_1_ and MC_2_ when the axial displacement of the TPU rotor’s shaft is present (**a**,**c**) and absent (**b**).

**Figure 7 sensors-21-03463-f007:**
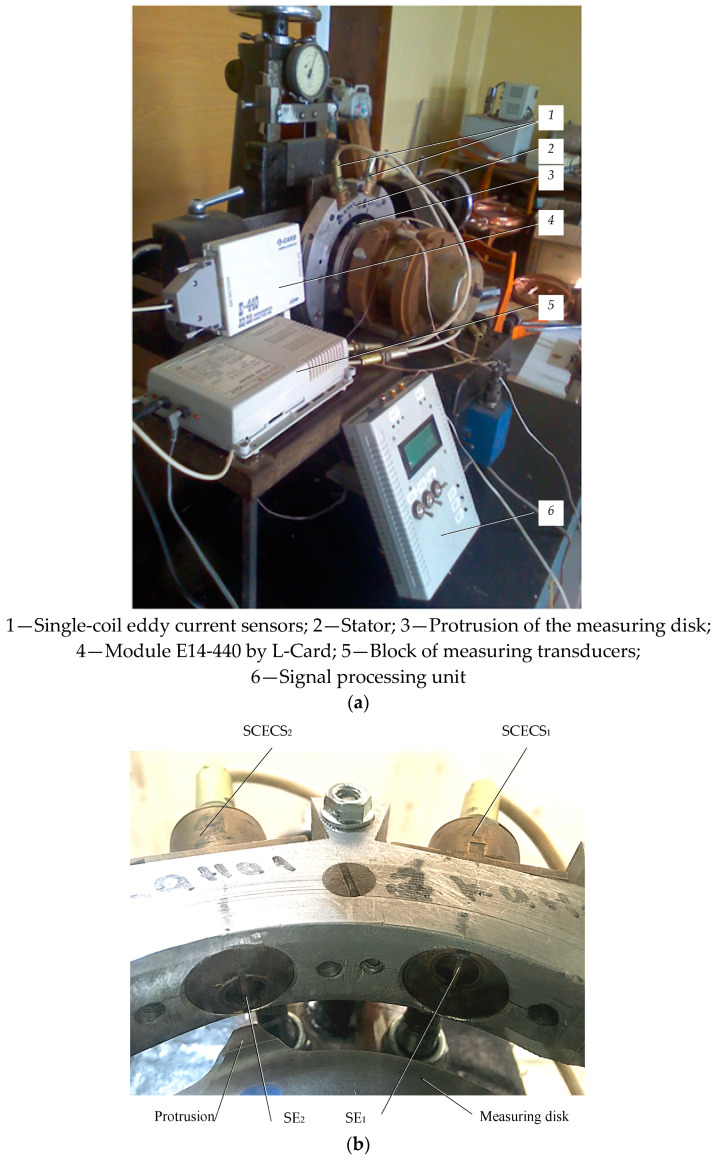
Laboratory bench for testing the feasibility of the proposed method (**a**) and the fragment of the stator with installed SCECS (**b**).

**Figure 8 sensors-21-03463-f008:**
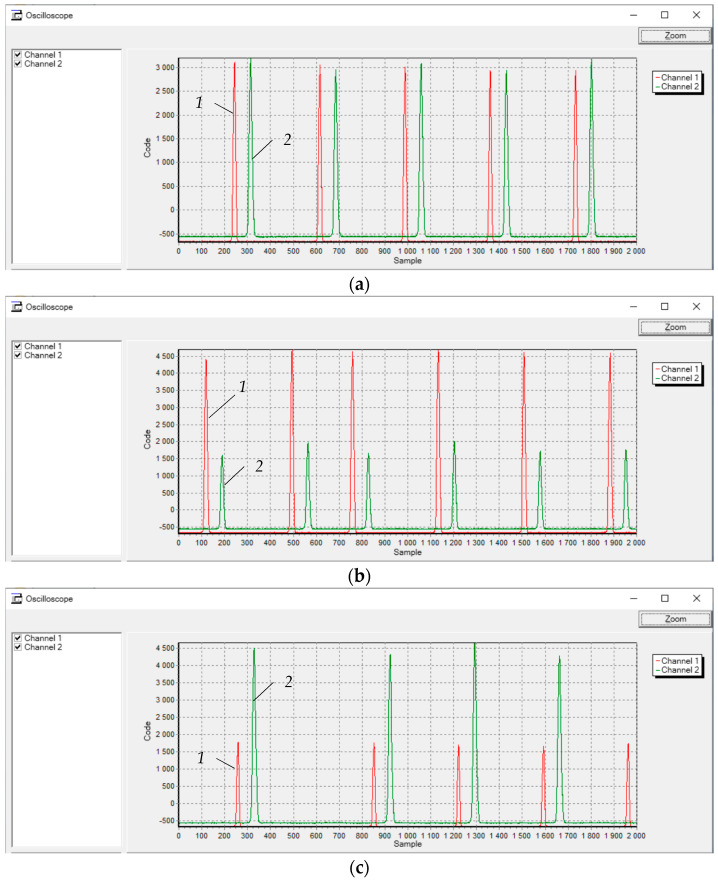
Time diagrams of changes in the output signals of the MC_1_ and MC_2_ when the axial displacement of the TPU rotor’s shaft is absent (**a**) and present (**b**,**c**).

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
