# Peer review of "Wear Diagnostics of the Thrust Bearing of NK-33 Turbo-Pump Unit on the Basis of Single-Coil Eddy Current Sensors"

_sensors, 2021, doi:10.3390/s21103463_

Round 1

Reviewer 1 Report

In my opinion the paper poses an interesting problem and offers a solid scientific solution. Furthermore, the topic covered in the paper is of great interest for the journal’s readers. However, I have some questions and/or suggestions for the authors:

  1. Have you considered the possibility of including any kind of automatic data processing instead of using some waveform parameters for early fault detection?
  2. The authors should consider extending the state of the art description, including some more international (not just Russian) references. In particular I suggest considering condition-based maintenance literature to cover this shortage of references.

Author Response

Dear Reviewer,

we wish to thank you for the constructive and precious comments. You acknowledged the potential of our work, although recognizing some important limits and gave us the chance to revise it. We agree with all your comments and our responses are given below.

Point 1: Have you considered the possibility of including any kind of automatic data processing instead of using some waveform parameters for early fault detection?

Response 1: Of course, the real operation conditions of the early fault detection system imply the automatic calculation of the shaft’s axial displacement in the CJTB. The «physical value» of the displacement is transferred to the monitoring unit for further analysis and decision making. Algorithms for coordinates calculating of such offset are well-known. We used them to determine the radial and axial displacements of the blades’ tips of gas turbine engines during bench tests of the power plants. The results were published earlier, for example in [24] (the version of the manuscript after review). Therefore, while preparing the article, we decided to provide only the waveforms as we considered them more suitable to the demonstration of the proposed approach feasibility. However, taking into account the interest in this issue, we have improved the text of the article and described in detail the essence of the algorithm for the clearance and axial displacements computing (Section 4: Page 10: Lines 370-379). We have also expanded the section on experimental validation of our approach by adding the corresponding results of the automatic calculation of the desired axial displacement of the shaft (Section 5: Page 13: Lines 522-541).

Point 2: The authors should consider extending the state of the art description, including some more international (not just Russian) references. In particular I suggest considering condition-based maintenance literature to cover this shortage of references.

Response 2: The authors thank you for these recommendations which will certainly improve our article. We have made appropriate additions to the text of the introduction, considering the most applied approaches for the diagnosis of the bearing mount assemblies (Page 2: Lines 54-68; references [6-15]). Additional references have been also made to international literature sources focusing on eddy current sensors used for aerospace applications (Page 2: Lines 72-73; references [19-22]).

We hope the manuscript is now suitable for publication.

Sincerely,
The authors.

Reviewer 2 Report

Dear Authors, 

The problem of early wear diagnostics of the combined journal-and-thrust bearing of the turbo-pump unit (TPU) of the liquid-propellant rocket engine NK-33 is considered in this manuscript. Regarding the first round review, the reviewer has the following comments:

  1. The contribution(s) is not clear.
  2. The paper organization is missed. Please add paper organization in the last paragraph of introduction.
  3. How we can verified the proposed method?
  4. Which type of sensor is used in this work?
  5. Explain Figure 1.b more clear.
  6. The results need more expand.

Regards, 

Author Response

Dear Reviewer,

we wish to thank you for the comments that allowed us to refine the content of the paper. We have carefully reviewed them and have revised the manuscript accordingly. Our responses are given in a point-by-point manner below.

Point 1: The contribution(s) is not clear.

Response 1: We used the recommendations in Instructions for Authors (https://www.mdpi.com/sensors/instructions) when specifying authors contributions in the short. We are grateful to the reviewer for the opportunity to elaborate each author’s contribution to the creation and writing of the article in detail.

The task of the developing the early fault detection system was formulated by V.D. He also defined the basic requirements and limitations of the system.

The discussions between S.B. and Y.S. led to the formation of the basic ideas of the early fault detection system. Later these ideas formed the foundation of the proposed method and the prototype of the system.

V.B., V.D. and Y.S. jointly developed the fault detection system’s hardware and the laboratory bench for testing the feasibility of the proposed method: V.D. ensured the creation of the TNA simulator; Y.S. developed the modified design of the SCECS with the offset SE and V.B. manufactured and adjusted the electronic units of the system’s prototype. V.B. also conducted the experiments with the prototype of the fault detection system and ensured the collection of initial data.

The processing and interpretation of the results were carried out jointly by S.B. and Y.S.

Writing the initial draft (including the translation) was made by S.B.

Further work on the article, its critical review, commentary and revision on pre-publication stage were carried out and are being carried out by S.B., V.D. and Y.S.

Point 2: The paper organization is missed. Please add paper organization in the last paragraph of introduction.

Response 2: The Instructions for Authors (https://www.mdpi.com/journal/sensors/ instructions) do not provide the paper organization in the introduction:

The introduction should briefly place the study in a broad context and highlight why it is important. It should define the purpose of the work and its significance, including specific hypotheses being tested. The current state of the research field should be reviewed carefully and key publications cited. Please highlight controversial and diverging hypotheses when necessary. Finally, briefly mention the main aim of the work and highlight the main conclusions. Keep the introduction comprehensible to scientists working outside the topic of the paper.

We therefore thought that a short reference to the content of the article in its original version (The description of the proposed approach and the results of its feasibility testing obtained on a laboratory prototype of the system for the early diagnostics of the CJTB wear of the TPU of NK-33 are given.”) would be quite sufficient for the readers. However, due to this comment we have decided to describe the content of the article in detail in the last paragraph of the introduction (Page 2: Lines 78-88). We also made additions to the text of the introduction, reviewing the most well-known approaches for the diagnosis of the bearing mount assemblies (Page 2: Lines 54-68; references [6-15]) and provided additional links to the international literature sources focusing on eddy current sensors used for aerospace applications (Page 2: Lines 72-73; references [19-22]).

Point 3: How we can verify the proposed method?

Response 3: Similar methods for the measuring of radial and axial displacements of the constructive elements of gas turbine engines were verified under real operating conditions during power plants’ bench tests where the performance, reliability and metrological characteristics of the methods were confirmed (see, for example, Borovik, S.; Sekisov, Y. Single-Coil Eddy Current Sensors and Their Application for Monitoring the Dangerous States of Gas-Turbine Engines. Sensors 2020, 20, 2107. https://doi.org/10.3390/s20072107). In these tests the radial clearance was an informational parameter, and the axial displacement was an interfering factor. In the case under consideration, on the contrary, axial displacement is an information parameter, and radial clearance is an interfering factor. In addition, the strict restrictions on changes in the design of the TPU body are the mandatory condition of the developers of the monitoring object. That is why a new sensor’s design was developed to support the proposed method. As for the rest, the approaches are identical.

To verify the proposed in the article method in laboratory conditions the similar to the described in section 5 bench should be assembled. In general, SCECS may be replaced by any other displacement sensors (eddy-current, capacitive, optical) on condition that they are sensitive to both types of the displacements of the protrusion tip. The positioning of sensors on the stator shell (displacement of the sensors’ SE) should be the same as in the article.

Point 4: Which type of sensor is used in this work?

Response 4: Perhaps we did not fully understand the question of the respected reviewer, but the sensor’s type is indicated in the title of the article. We used the single-coil eddy current sensors (SCECS). Section 3 is devoted to the description of them. SCECS are a separate and independent branch among the eddy current sensors used in aerospace engine building (Page 2: Lines 72-73). For the sake of clarity, we have added a figure with a fragment of the stator simulator with the installed SCECS (Section 5: Page 11: Figure 7b).

More information about SCECS can be fined in:

  1. Borovik, S.; Sekisov, Y. Single-Coil Eddy Current Sensors and Their Application for Monitoring the Dangerous States of Gas-Turbine Engines. Sensors 2020, 20, 2107. https://doi.org/10.3390/s20072107
  2. https://encyclopedia.pub/1002

Point 5: Explain Figure 1.b more clear.

Response 5: The description of Figure 1b was changed and detailed (Section 2: Page 3: Lines 128-151).

Point 6: The results need more expand.

Response 6: We have expanded section 5 with experimental results by adding the figure with a fragment of the stator simulator with installed SCECS (Section 5: Page 11: Figure 7b) and the results of the automatic calculation of the desired axial displacement of the measuring disk (Section 5: Page 13: Lines 522-541).

We hope the manuscript is now suitable for publication.

Sincerely,
The authors

Round 2

Reviewer 2 Report

Dear Authors,

Regarding the 2nd round review of the manuscript and response letter, it can be accepted for further processing.

Regards